# Pediatric overweight and obesity increased in Israel during the COVID-19 period

**Adam Rose**[1,2]*, **Eliana Ein Mor**[1,2], **Michal Krieger**[1,2], **Arie Ben-Yehuda**[2,3],
**Shoshana Revel-Vilk**[2,4,5], **Arnon D. Cohen**[2,6], **Eran Matz**[2,7], **Edna Bar-Ratson**[2,8],
**Ronen Bareket**[2,9,10], **Ora Paltiel**[1,2,3], **Ronit Calderon-Margalit**[1,2]

**1** Braun School of Public Health and Community Medicine, Hadassah Medical Organization, Faculty of
Medicine, Hebrew University of Jerusalem, Jerusalem, Israel, **2** National Program for Quality Indicators in
Community Healthcare in Israel, Jerusalem, Israel, **3** Hadassah Medical Organization, Jerusalem, Israel,
**4** Pediatric Hematology/Oncology Unit, Department of Pediatrics, Shaare Zedek Medical Center, Jerusalem,
Israel, **5** Faculty of Medicine, Hebrew University, Jerusalem, Israel, **6** Clalit Health Services, Tel Aviv, Israel,
**7** Leumit Health Services, Tel Aviv, Israel, **8** Maccabi Health Services, Tel Aviv, Israel, **9** Meuhedet Health
Services, Tel Aviv, Israel, **10** Department of Family Medicine, Sackler School of Medicine, Tel Aviv
University, Tel Aviv, Israel

* adam.rose@mail.huji.ac.il

Koningin Fabiola: Hopital Universitaire des Enfants
Reine Fabiola, BELGIUM

**Data Availability Statement:** QICH data are
collected by agreement from Israel's four health
maintenance organizations (HMOs). The use of
these data is governed by the data sharing

## Abstract

Reports from many settings suggest that pediatric overweight and obesity increased in 2020
and 2021, presumably due to lifestyle changes associated with the COVID-19 pandemic.
Many of these previous reports have relied on convenience samples or subsets of the popu-
lation. Here, we present results of a longitudinal study of the entire population of Israel, a
nation of approximately 9 million people, with the proportion with underweight, normal
weight, overweight, and obesity at age 7 and at age 14–15, across the years 2017–2021.
Our results show that the prevalence of overweight and obesity, which had been steady or
improving through 2019, increased relatively quickly in 2020 and 2021. For example, among
7-year-olds, the percentage of children with obesity in 2019 was 6.8% (99% CI: 6.69–7.05),
and by 2021, it had increased to 7.7% (99% CI: 7.53–7.93). There were important disparities
in overweight and obesity based on SES; for example, the rate ratio for obesity comparing
the poorest with the wealthiest 14–15-year-olds in 2019 was 1.63 (99% CI: 1.55–1.72).
However, these disparities did not change meaningfully in 2020 and 2021, implying that
while obesity did become more prevalent, this increase in prevalence was not differential
across socioeconomic status. Like many other nations, Israel too experienced considerable
increases in pediatric overweight and obesity in 2020–2021, erasing the improvements of
the previous years among younger children.

## 1. Introduction

The prevalence of pediatric overweight and obesity has increased over the past two decades,
especially in high income countries [1]. Since children with obesity are more likely to become
adults with obesity [2], addressing pediatric overweight and obesity is a major public health
goal [3, 4]. Indeed, limited weight gain early in life may be associated with better health

agreement that allows the data to be collected in one place. One of the stipulations of this data sharing agreement is that the data must remain on a particular encrypted server located at Hebrew University, and must be analyzed from that location, and cannot be shared with outside parties. It is possible for outside researchers to apply to use the data in conjunction with the QICH Steering Committee. Analyses would be conducted by QICH staff, and the results would be sent to the collaborators, because the data cannot leave the server. Requests to work with QICH data should be sent to the QICH Steering Committee at contact@israelhealthindicators.org.

**Funding:** The author(s) received no specific funding for this work.

**Competing interests:** The authors have declared that no competing interests exist.

throughout the lifespan, by reducing the prevalence of obesity during adulthood and the associated burden of non-communicable diseases [5, 6]. However, the effort to limit overweight and obesity during childhood is a battle that many societies have been losing over the past several decades.

Beginning in 2020, due to the COVID-19 pandemic, the entire world experienced disruptions to daily routines of life. Many aspects of COVID-19, especially lockdowns and quarantines, contributed to an obesogenic lifestyle, such as reduced opportunities for physical activity and changes in dietary patterns [7–9]. Reports from many nations and many settings have suggested that pediatric overweight and obesity did increase during the peak years for COVID-19, 2020 and 2021 [7–19]. However, many of these reports have used convenience samples. Thus, there is a need to examine increases in pediatric overweight and obesity during the COVID-19 period using population-representative databases.

We therefore used a database that represents the entire population of Israel, a nation with a population of approximately 9.4 million people. Israel had three lockdown periods for COVID-19, each lasting about one month (March-April 2020, September 2020 and December 2020-January 2021). However, school closures extended beyond these lockdowns, and as a result, the amount of school missed by Israeli students was well above the OECD average [20]. This disruption of children's normal routines, combined with more time spent using a computer, had considerable potential to promote weight gain through changes in patterns of eating and activity [7–9]. The Israeli National Program for Quality Indicators in Community Healthcare (QICH) has collected data on weight distribution of 7-year-old children since 2013, and of 14–15-year-old children since 2017. While we had seen a halt to the rising rate of obesity among Israeli 7-year-olds since 2015, and even perhaps a slight reduction in obesity, we were concerned that the COVID-19 pandemic might have reversed this recent progress. Our main study questions were: did we see an increase in pediatric overweight and obesity in 2020 and 2021 compared to previous years, and were these changes worse among children of lower SES?

## 2. Materials and methods

### 2.1. Data source

In the State of Israel, all citizens and permanent residents are members of one of the four health maintenance organizations (HMOs) that supply health services in the community [21]. All HMOs support and cooperate with QICH program, in development, assessment, providing the national indicators, data, and publication of the quality indicators. The data represent the entire Israeli civilian population. QICH data are aggregated by sex, age and socioeconomic status, and therefore they are de-identified and cannot be re-identified. These data are used for quality assurance purposes, and are the basis of annual public reports.

Because these data are de-identified according to the strictest definition, this study was determined to be not human subjects research and therefore exempt from ethics review, per the Helsinki Committee of Hadassah Medical Organization (dated June 14, 2023). In addition, the Helsinki Committee waived the requirement for informed consent, for the same reason. The study authors did not, at any time, have access to information that could be used to identify individual study participants.

### 2.2. Dataset

QICH data are organized by calendar year. Therefore, in this manuscript, we will be comparing the years 2020 and 2021, coinciding with the COVID-19 pandemic, with the years 2017–2019, which were prior to the pandemic. The analyses reported here were conducted between September 2022-May 2023.

## 2.3. Measures

This manuscript will focus on two measures, which are part of the QICH quality measure set for Israel. The first measure is the proportion of children who have height and weight recorded, sufficient to calculate body mass index (BMI). Children are assessed for weight and height (in order to calculate BMI) at the age of 7 and at the age of 14–15. These assessments are carried out by a nurse or a physician at regular office visit.

The second measure Is the proportion of children who are in different weight categories. For this measure, the denominator consists of those children who have been assessed for both weight and height. The BMI categories are calculated according to the WHO growth charts [22], meaning that Israel's children are assessed compared to the international data underlying the WHO curves:

- Children with Underweight (BMI $\leq$2.3 percentile, z-score below -2)

- Children with Normal Weight (BMI between 2.3–85 percentile)

- Children with Overweight (BMI between 85–97.7 percentile, z-score between 1–2)

- Children with Obesity (BMI above 97.7 percentile, z-score above 2)

## 2.4. Covariates

Covariates included age (age 7 or age 14–15), sex, and area-level socioeconomic status (SES), as reported by the POINTS company [23]. POINTS divides Israeli residential locations into 10 levels of SES; we reduced those to 4 levels here, by combining levels 1–3 (very poor), 4–5 (moderately poor), 6–7 (middle income), and 8–10 (wealthy).

## 2.5. Analyses

We examined both measures by year, namely 1) the proportion of children who had a BMI measurement, and 2) the proportion of those measured who were in different BMI categories. We report these measures separately for age 7 and for age 14–15. We also stratified the population into SES groups to compare time trends across these groups, and present the rate ratio (RR) for the difference between lowest and highest SES. In addition, we present the data stratified by all three covariates (age, sex, and SES). We calculated a 99% Confidence Interval (99% CI) for all point estimates and rate ratios, as well as p-values for linear trend over the period from 2017–2021. All analyses were performed using R studio v 2022.07.2+576 and Microsoft Excel.

## 3. Results

### 3.1. Characteristics of the population

Table 1 shows basic characteristics of the included population as of 2021 (165,274 children aged 7 years, and 285,292 children aged 14–15 years). Both age groups were 51% male, and the social class distribution was similar between the age groups, with 26% and 27.5% of children aged 7 and 14–15, respectively, in the poorest SES group and 19% and 20%, respectively, in the wealthiest group.

**Table 1. Characteristics of those included in the study during the year 2021.** Column percentages are given.

|  | **Age 7** | **Age 14–15** |
|---|---|---|
| Entire Sample | 165,274 (100%) | 285,292 (100%) |
| Boys | 85,020 (51.4%) | 146,594 (51.4%) |
| Girls | 80,254 (48.6%) | 138,698 (48.6%) |
| SES Strata |  |  |
| 1–3 (poorest) | 41,542 (26.0%) | 76,194 (27.5%) |
| 4–5 | 44,372 (27.7%) | 73,932 (26.7%) |
| 6–7 | 43,917 (27.4%) | 71,485 (25.8%) |
| 8–10 (wealthiest) | 21,603 (18.9%) | 55,352 (20.0%) |

### 3.1.1. First measure: Proportion of children assessed for height and weight

In Table 2, we show the proportion of children age 7 and age 14–15 who recorded height and weight information, sufficient to calculate BMI. Documentation of these parameters was consistent across the pre-COVID period from 2017–2019. In 2017, the proportion with both height and weight recorded was 79.7% among 7-year-olds (99% CI: 79.4%-79.9%), and 79.4% among 14-15-year-olds (99% CI: 79.2%-79.6%). In 2020 and 2021, the levels of documentation were lower. In 2021, there was 71.7% documentation among 7-year-olds (99% CI: 73.5%-73.9%), and 73.7% documentation among 14-15-year-olds (99% CI: 73.5%-73.9%). The p-value for linear trend over time, suggesting decreased documentation in later years, was statistically significant in both age groups.

Stratification by SES groups (Table 2) showed that documentation was roughly equal across SES groups among 7-year-olds (rate ratio of poorest vs. wealthiest in 2021, 1.00). However, documentation was slightly better among higher SES groups among 14–15-year-olds (rate ratio of poorest vs. wealthiest in 2021, 0.94, 99% CI: 0.93–0.95), a disparity that had worsened since 2019 (when the rate ratio for documentation was 0.99 (99% CI 0.98–1.00).

**Table 2. Documentation of BMI information documentation by year, overall and stratified by SES categories.** Percentages are given.

|  |  | **2017** | **2018** | **2019** | **2020** | **2021** |
|---|---|---|---|---|---|---|
| Age 7 | Total n | 156,712 | 156,355 | 160,187 | 161,253 | 165,274 |
|  | Overall Documentation Rate | 79.7 | 79.2 | 80.9 | 77.4 | 71.7 |
| Age 7, stratified | SES 1–3 (poorest) | 81.8 | 80.2 | 82.6 | 79.5 | 71.7 |
|  | SES 4–5 | 79.2 | 78.6 | 79.4 | 77.4 | 70.4 |
|  | SES 6–7 | 78.3 | 78.6 | 77.4 | 76.8 | 71.8 |
|  | SES 8–10 (wealthiest) | 75.6 | 75.7 | 76.9 | 76.3 | 71.5 |
|  | Rate Ratio (poorest compared to wealthiest) | 1.08 | 1.06 | 1.07 | 1.04 | 1.00 |
|  | 99% CI for Rate Ratio | 1.07–1.09 | 1.05–1.07 | 1.06–1.09 | 1.03–1.05 | 0.99–1.01 |
| Age 14–15 | Total n | 270,646 | 278,323 | 281,917 | 282,431 | 285,292 |
|  | Overall Documentation Rate | 79.4 | 78.7 | 79.9 | 75.8 | 73.7 |
| Age 14–15, stratified | SES 1–3 (poorest) | 76.7 | 76.5 | 77.4 | 73.5 | 70.4 |
|  | SES 4–5 | 79.7 | 79.2 | 80.0 | 76.9 | 73.9 |
|  | SES 6–7 | 81.0 | 79.8 | 79.6 | 77.0 | 75.7 |
|  | SES 8–10 (wealthiest) | 79.8 | 78.5 | 78.6 | 76.3 | 75.0 |
|  | Rate Ratio (poorest compared to wealthiest) | 0.96 | 0.97 | 0.99 | 0.96 | 0.94 |
|  | 99% CI for Rate Ratio | 0.96–0.97 | 0.97–0.98 | 0.98–1.00 | 0.96–0.97 | 0.93–0.95 |

### 3.1.2. Second measure: Proportion of children in different weight categories

In Table 3, we show the proportion of children in different weight categories–among those who recorded sufficient information to calculate BMI. Among 7-year-olds, across the pre-COVID years, the proportion of children with obesity was decreasing, reaching a low of 6.8% in 2019 (99% CI: 6.6%-7.0%). By 2021, it had increased again to 7.7% (99% CI: 7.5%-7.9%), a level not seen in our data previously. The rate of children with overweight also worsened; it had decreased to 10.9% in 2019 (99% CI: 10.7%-11.2%), but then had climbed back up to 11.9% by 2021 (99% CI: 11.6%-12.2%), reversing a previous trend of improvement over time.

We also examined these results for 7-year-olds, stratified by SES groups, and focusing on obesity (Fig 1). A full presentation of the same results, including all four BMI categories and including rate ratios, is in Table 4. In each SES group, we see that the percentage of children with obesity was relatively steady over the first several years, but increased in 2021. The increase in obesity seemed to impact lower-SES groups disproportionately, although this difference did not reach statistical significance, as the confidence intervals overlapped. For example, the rate ratio between the lowest and highest-SES groups in 2019 was 1.08 (99% CI: 0.99–1.19); in 2020, it increased to 1.18 (99% CI: 1.08–1.29).

Among 14–15-year-olds overall (Table 3), the rate of obesity was 10.2% in 2017. This percentage increased every year thereafter, reaching 11.8% in 2021 (p for linear trend = 0.002)–again, a level not seen in our data previously. The rate of overweight, which had held steady at 17.9% or 17.8% throughout 2017–2019, also increased during 2020–2021, reaching a high of 18.5% in 2021 (p for linear trend = 0.002).

We also stratified these results for 14–15-year-olds by SES groups. Fig 2 focuses on obesity only, while Table 5 presents full results including all body weight categories and rate ratios. Again, we see that the proportion of children with obesity was lowest in 2017, and increased over the entire period (p for linear trend = 0.002). In this age group, while poorer children

**Table 3. BMI categories by year, by age.** Percentages are given.

| | 2017 | 2018 | 2019 | 2020 | 2021 | P value for trend over time |
|---|---|---|---|---|---|---|
| **At age 7** | | | | | | |
| Underweight (BMI ≤ 2.3%) | 4.5 | 4.3 | 4.5 | 4.3 | 4.3 | 0.31 |
| 99% CI | 4.32–4.62 | 4.18–4.48 | 4.40–4.70 | 4.16–4.45 | 4.12–4.42 | |
| Normal (BMI 2.3–85%) | 76.5 | 77.6 | 77.7 | 77.5 | 76.1 | 0.75 |
| 99% CI | 76.2–76.8 | 77.3–77.8 | 77.3–77.9 | 77.2–77.9 | 75.8–76.4 | |
| Overweight (BMI 85–97.7%) | 11.9 | 11.2 | 10.9 | 11.3 | 11.9 | 0.96 |
| 99% CI | 11.7–12.2 | 11.0–11.4 | 10.7–11.2 | 11.1–11.6 | 11.6–12.2 | |
| Obese (BMI ≥ 97.7%) | 7.1 | 6.9 | 6.8 | 6.8 | 7.7 | 0.56 |
| 99% CI | 6.9–7.3 | 6.72–7.09 | 6.69–7.05 | 6.62–7.00 | 7.53–7.93 | |
| **At age 14–15** (total n) | | | | | | |
| Underweight (BMI ≤ 2.3%) | 3.4 | 3.6 | 3.5 | 3.6 | 3.7 | 0.04 |
| | 3.27–3.48 | 3.45–3.65 | 3.42–3.62 | 3.50–3.71 | 3.58–3.80 | |
| Normal (BMI 2.3–85%) | 68.6 | 68.1 | 67.8 | 67.1 | 66.0 | 0.002 |
| 99% CI | 68.3–68.8 | 67.8–68.3 | 67.6–68.1 | 66.8–67.3 | 65.7–66.3 | |
| Overweight (BMI 85–97.7%) | 17.9 | 17.8 | 17.8 | 18.2 | 18.5 | 0.04 |
| 99% CI | 17.7–18.1 | 17.6–18.1 | 17.6–18.0 | 18.0–18.4 | 18.3–18.7 | |
| Obese (BMI ≥ 97.7%) | 10.2 | 10.5 | 10.9 | 11.1 | 11.8 | 0.002 |
| 99% CI | 10.0–10.3 | 10.3–10.7 | 10.7–11.0 | 11.0–11.3 | 11.7–1.02 | |

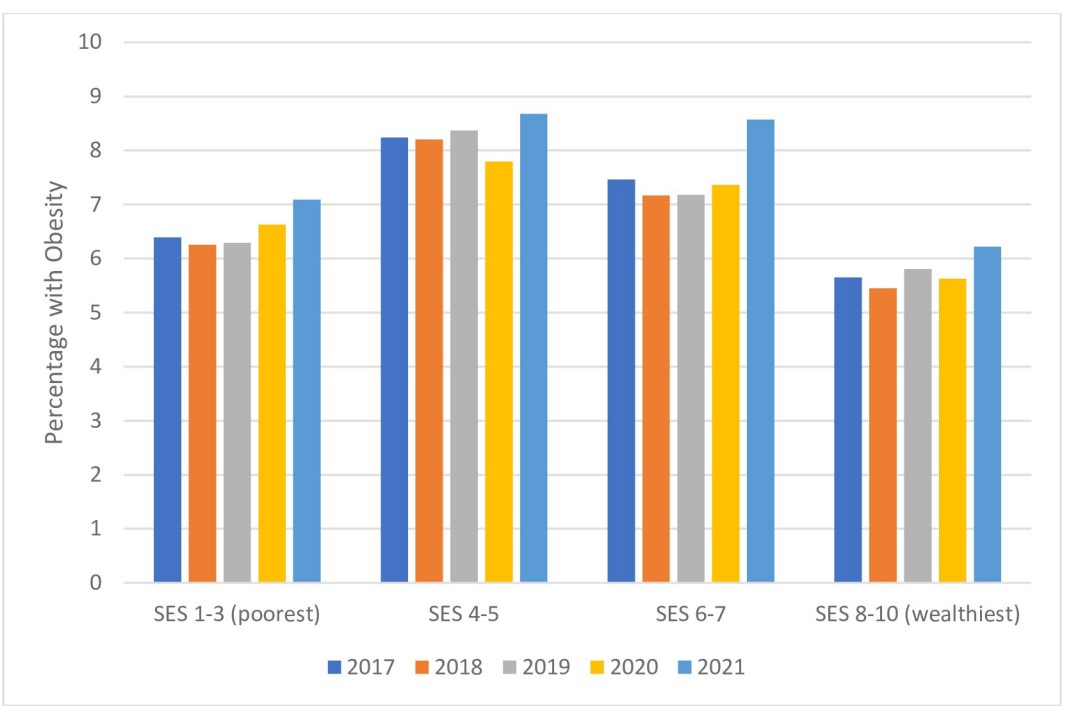

**Fig 1. Percent prevalence of obesity by year among 7-year-old children, stratified into four SES groups.**

certainly had a higher rate of obesity, there is no clear indication that this disparity worsened over time (rate ratio of poorest to wealthiest groups, 1.68 in 2017 and 1.66 in 2021).

In Table 6, we show obesity rates, stratified by both SES and by sex. We see that teenage boys from the lowest SES group have the highest prevalence of obesity, as many as 15.9% of them in 2021, compared to 10.7% among girls. The rate of obesity in the next poorest SES stratum is also high for boys, 15.3%, compared to girls at 11.9%. Socioeconomic disparities in obesity also changed differently among teenage boys and among teenage girls during the COVID period. Among boys, the rate ratio of obesity, comparing the poorest SES group to the wealthiest, began at 1.72 in 2017, improved slightly to 1.68 in 2019, and then worsened to 1.86 in 2021, although this change did not reach statistical significance (p for linear trend = 0.07). Among girls, the socioeconomic disparity actually decreased every year, falling from a rate ratio of 1.61 in 2017 to as low as 1.42 in 2021 (p for linear trend = 0.004).

## 4. Discussion

This report is based on Israel's national program of quality indicators in community healthcare, and includes the entire population of Israel. We examined whether the prevalence of pediatric overweight and obesity increased during the COVID-19 period, namely in 2020 and 2021, compared to the previous three years (2017–2019). In 2020 and 2021, we saw a decrease in BMI documentation at ages 7 and 14–15. This probably reflects decreased in-person visits with the family doctor or pediatrician during the COVID period, with a shift to virtual visits instead, during which weight and height cannot be measured. We also saw increased obesity rates in both age groups specifically among boys, and especially among the 14–15-year-olds, which revered a previous trend of improvement among younger children and accelerated a trend of worsening obesity among the older children. As other studies have shown, the COVID-19 period was associated with obesogenic changes in diet and lifestyle [7–9], which

**Table 4. Percent BMI categories by year, stratified into four SES groups–age 7.**

| Underweight | 2017 | 2018 | 2019 | 2020 | 2021 | P value for trend over time |
|---|---|---|---|---|---|---|
| 1–3 (poorest) | 6.3 | 6.1 | 7.0 | 6.2 | 6.2 | |
| 4–5 | 4.0 | 4.0 | 3.8 | 3.9 | 4.1 | |
| 6–7 | 3.4 | 3.5 | 3.6 | 3.5 | 3.2 | |
| 8–10 (wealthiest) | 3.1 | 2.9 | 3.2 | 3.2 | 2.9 | |
| rate ratio (poorest/wealthiest) | 2.06 | 2.10 | 2.19 | 1.92 | 2.12 | 0.04 |
| 99% CI | 1.82–2.33 | 1.85–2.38 | 1.97–2.45 | 1.72–2.14 | 1.89–2.38 | |
| Normal Weight | | | | | | |
| 1–3 (poorest) | 76.3 | 77.6 | 77.1 | 76.8 | 76.1 | |
| 4–5 | 75.3 | 75.9 | 76.2 | 76.3 | 74.6 | |
| 6–7 | 76.6 | 77.6 | 77.7 | 77.4 | 75.7 | |
| 8–10 (wealthiest) | 79.1 | 79.9 | 79.9 | 79.9 | 78.6 | |
| rate ratio (poorest/wealthiest) | 0.96 | 0.97 | 0.96 | 0.96 | 0.97 | 0.64 |
| 99% CI | 0.96–0.98 | 0.96–0.98 | 0.95–0.98 | 0.95–0.97 | 0.96–0.98 | |
| Overweight | | | | | | |
| 1–3 (poorest) | 11.0 | 10.0 | 9.7 | 10.4 | 10.7 | |
| 4–5 | 12.5 | 11.9 | 11.6 | 12.0 | 12.7 | |
| 6–7 | 12.5 | 11.8 | 11.5 | 11.8 | 12.5 | |
| 8–10 (wealthiest) | 12.2 | 11.7 | 11.1 | 11.3 | 12.3 | |
| rate ratio (poorest/wealthiest) | 0.90 | 0.86 | 0.87 | 0.92 | 0.87 | 0.99 |
| 99% CI | 0.85–0.97 | 0.80–0.92 | 0.81–0.93 | 0.86–0.98 | 0.81–0.93 | |
| Obesity | | | | | | |
| 1–3 (poorest) | 6.4 | 6.3 | 6.3 | 6.6 | 7.1 | |
| 4–5 | 8.2 | 8.2 | 8.4 | 7.8 | 8.7 | |
| 6–7 | 7.5 | 7.2 | 7.2 | 7.4 | 8.6 | |
| 8–10 (wealthiest) | 5.7 | 5.5 | 5.8 | 5.6 | 6.2 | |
| rate ratio (poorest/wealthiest) | 1.13 | 1.15 | 1.08 | 1.18 | 1.14 | 0.73 |
| 99% CI | 1.02–1.24 | 1.04–1.27 | 0.99–1.19 | 1.08–1.29 | 1.04–1.24 | |

likely accounts for these changes here in Israel, as it did in so many other places. Given the known relationship between childhood obesity and adult obesity and health outcomes [2, 5, 6], these changes have the potential to impact long-term public health in Israel for the worse.

Our SES and sex-stratified results are among the most interesting findings here. Specifically, rates of obesity, and socioeconomic disparities in obesity, were much worse among 14–15-year-old boys than among girls of the same age. Moreover, this socioeconomic disparity worsened among boys during the COVID period and actually improved among girls. This could be due to more social pressure among girls of this age to avoid gaining weight than among boys, an effect which has been shown to be enhanced by more exposure to online settings such as Zoom or social media [24]. In addition, there could have potentially been differential patterns of physical activity by sex during the pandemic period. This result also reminds us that underneath the average results (i.e., both sexes combined), there may be subgroups that are moving in different directions if only one looks carefully. These results also imply a need to target obesity prevention among this age group specifically to boys.

There have been numerous recent reports about childhood obesity rates during the COVID period. These reports have used various methods and have originated from various locations [7, 8, 10, 13, 14, 17]. Two related reports about children treated in Kaiser Permanente of Southern California found increasing rates of overweight and obesity in that population during the

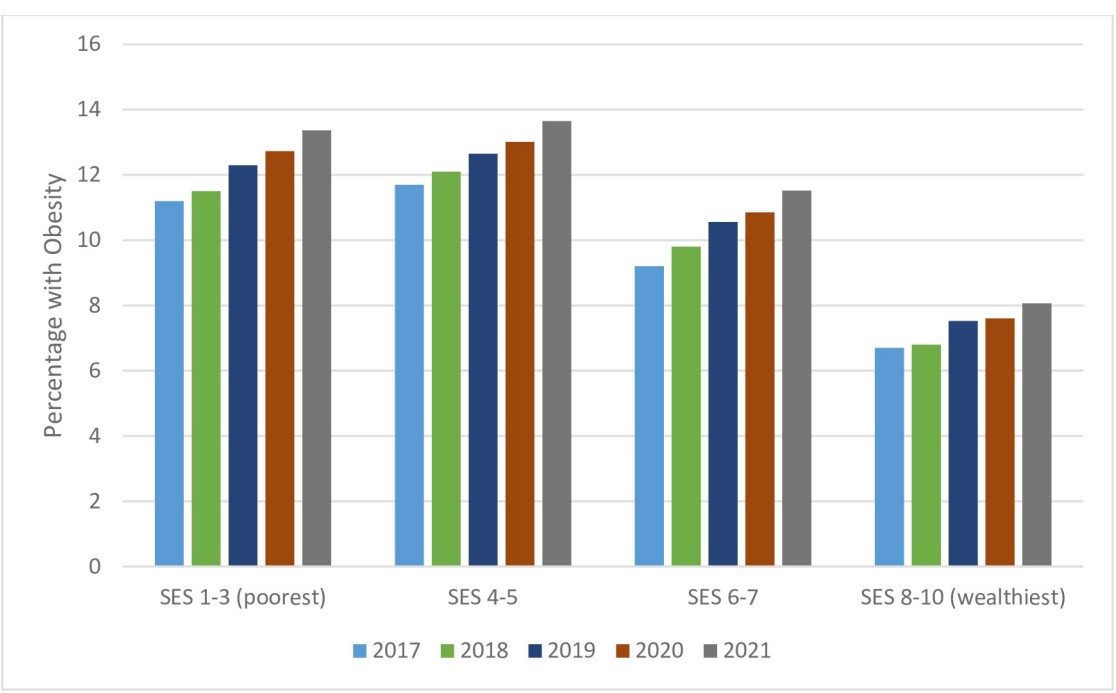

**Fig 2. Percent prevalence of obesity by year among 14-15-year-old children, stratified into four SES groups.**

COVID period. In their study, BMI increased by 1.72 for Hispanic youths, and 1.70 for Black youths, and only by 1.16 for White youths [12, 18]. A longitudinal study of children treated by three large health systems in Massachusetts also showed increasing rates of overweight and obesity, again, with greater increases in obesity among minority populations, especially Black and Hispanic children [19]. A large, annual web-based survey in South Korea, which relied on self-reported height and weight, also found an increasing rate of obesity over this period, increasing from 11.0% to 12.1% [9]. In a study that echoes our findings, a school-based program in Isparta Province, Turkey, showed that improvements in the rates of overweight and obesity that had occurred since 2005 were suddenly reversed, with 2021 numbers being higher than any they had previously observed [16]. A study relying on a large, common electronic medical record in the USA found that BMI increased very quickly during the early pandemic, but then stabilized or even improved slightly later in the pandemic [15]. A probabilistic annual national sample from Peru showed that the proportion of children with obesity had increased from 6.4% to 7.8% between 2019 and 2021 [11].

Our study had important strengths. While some previous studies have relied on convenience samples, such as querying patients treated in a particular health system, others reflect a probability-based sample of a national population [9, 11]. However, ours is the first study of which we are aware that directly reflects the population of an entire country. In addition, we used a dataset that allowed year-by-year comparisons across three pre-COVID years (2017–2019) and two years impacted by COVID (2020–2021). However, our study also has limitations. Our data are aggregate, and therefore, certain types of individual-level analyses were not possible with these data. For example, it would not be possible to perform sensitivity analyses with different cutoffs for the BMI categories using these data.

Another possible limitation is that the decreases in documentation that we observed during the COVID period could have contributed to the apparent worsening of overweight and obesity during the same period. Indeed, our dataset is representative of the entire population of

**Table 5. Percent BMI categories by year, stratified into four SES groups–age 14–15.**

|  | 2017 | 2018 | 2019 | 2020 | 2021 | P value for trend over time |
|---|---|---|---|---|---|---|
| **Underweight** |  |  |  |  |  |  |
| 1–3 (poorest) | 3.4 | 3.7 | 3.5 | 3.5 | 3.6 |  |
| 4–5 | 3.6 | 3.8 | 3.6 | 3.7 | 3.7 |  |
| 6–7 | 3.3 | 3.4 | 3.5 | 3.6 | 3.8 |  |
| 8–10 (wealthiest) | 2.9 | 3.2 | 3.3 | 3.5 | 3.6 |  |
| rate ratio (poorest/wealthiest) | 1.18 | 1.17 | 1.07 | 1.01 | 1.00 | 0.005 |
| 99% CI | 1.08–1.33 | 1.08–1.31 | 0.99–1.19 | 0.93–1.12 | 0.92–1.10 |  |
| **Normal Weight** |  |  |  |  |  |  |
| 1–3 (poorest) | 67.2 | 66.7 | 66.4 | 65.2 | 63.6 |  |
| 4–5 | 66.1 | 65.4 | 65.5 | 64.2 | 63.6 |  |
| 6–7 | 69.5 | 68.7 | 67.6 | 66.9 | 66.0 |  |
| 8–10 (wealthiest) | 74.1 | 74.1 | 73.0 | 72.6 | 72.0 |  |
| rate ratio (poorest/wealthiest) | 0.91 | 0.90 | 0.91 | 0.90 | 0.88 | 0.06 |
| 99% CI | 0.90–0.92 | 0.89–0.91 | 0.90–0.92 | 0.89–0.91 | 0.87–0.89 |  |
| **Overweight** |  |  |  |  |  |  |
| 1–3 (poorest) | 18.1 | 18.1 | 17.8 | 18.5 | 19.4 |  |
| 4–5 | 18.6 | 18.7 | 18.3 | 19.1 | 19.0 |  |
| 6–7 | 17.9 | 18.0 | 18.4 | 18.6 | 18.7 |  |
| 8–10 (wealthiest) | 16.3 | 15.9 | 16.2 | 16.2 | 16.3 |  |
| rate ratio (poorest/wealthiest) | 1.11 | 1.14 | 1.10 | 1.14 | 1.19 | 0.17 |
| 99% CI | 1.07–1.16 | 1.09–1.18 | 1.06–1.14 | 1.10–1.19 | 1.15–1.23 |  |
| **Obese** |  |  |  |  |  |  |
| 1–3 (poorest) | 11.2 | 11.5 | 12.3 | 12.7 | 13.4 |  |
| 4–5 | 11.7 | 12.1 | 12.6 | 13.0 | 13.7 |  |
| 6–7 | 9.3 | 9.8 | 10.6 | 10.9 | 11.5 |  |
| 8–10 (wealthiest) | 6.7 | 6.8 | 7.5 | 7.6 | 8.1 |  |
| rate ratio (poorest/wealthiest) | 1.68 | 1.69 | 1.63 | 1.67 | 1.66 | 0.49 |
| 99% CI | 1.58–1.79 | 1.59–1.80 | 1.55–1.72 | 1.59–1.77 | 1.57–1.74 |  |

the State of Israel in that all citizens and legal residents receive care from the four Israeli HMO's; however, there were certainly some children in the population who did not see a physician during this time. For this reason, we present confidence intervals in this paper, because our sample, while extremely representative, may not include every single person living in the State of Israel. Relatedly, differential inclusion in the dataset may have impacted our study. If the children that physically attended visits were different from those who used telemedicine, this could have biased our estimates of obesity. In particular, it is possible that children of low SES, or their parents, lacked the computer equipment to conduct a virtual visit, or were less aware that it was available. However, our sex and SES-stratified results, presented in Table 6, do not support this idea. We saw that socioeconomic disparities in obesity among 14-15-year-olds worsened among boys, but improved among girls. It does not seem plausible that the families of girls had better access to computers and telemedicine than those of boys. Another point is that the documentation of height and weight decreased much more markedly among 14-15-year-olds than among 7-year-olds, and yet similar increases in obesity were observed in both age groups. Therefore, the increased prevalence of overweight and obesity that we observed during 2020 and 2021 would appear to be real changes, and not an artifact due to differential changes in documentation.

**Table 6. Obesity at ages 7 and 14–15 years by year, stratified by SES and by sex.** Percentages are given, as well as rate ratios.

| | All | | | | | | Boys | | | | | | Girls | | | | | |
|---|---|---|---|---|---|---|---|---|---|---|---|---|---|---|---|---|---|---|
| | 2017 | 2018 | 2019 | 2020 | 2021 | P value for trend | 2017 | 2018 | 2019 | 2020 | 2021 | P value for trend | 2017 | 2018 | 2019 | 2020 | 2021 | P value for trend |
| **Age 7** | | | | | | | | | | | | | | | | | | |
| 1–3 (poorest) | 6.4 | 6.3 | 6.3 | 6.6 | 7.1 | | 7.0 | 6.8 | 6.7 | 7.1 | 7.7 | | 5.8 | 5.6 | 5.8 | 6.2 | 6.5 | |
| 4–5 | 8.2 | 8.2 | 8.4 | 7.8 | 8.7 | | 8.6 | 8.6 | 8.5 | 8.0 | 9.1 | | 7.8 | 7.8 | 8.2 | 7.6 | 8.2 | |
| 6–7 | 7.5 | 7.2 | 7.2 | 7.4 | 8.6 | | 7.7 | 7.3 | 7.4 | 7.5 | 8.9 | | 7.2 | 7.0 | 7.0 | 7.2 | 8.2 | |
| 8–10 (wealthiest) | 5.7 | 5.5 | 5.8 | 5.6 | 6.2 | | 5.7 | 5.8 | 6.0 | 5.7 | 6.5 | | 5.6 | 5.0 | 5.6 | 5.6 | 5.9 | |
| Rate Ratio Poorest/ Wealthiest | 1.13 | 1.15 | 1.08 | 1.18 | 1.14 | 0.73 | 1.22 | 1.17 | 1.12 | 1.24 | 1.17 | 0.87 | 1.03 | 1.12 | 1.04 | 1.10 | 1.10 | 0.42 |
| 99% CI | 1.02–1.24 | 1.04–1.27 | 0.99–1.19 | 1.08–1.29 | 1.04–1.24 | | 1.07–1.39 | 1.03–1.34 | 0.99–0.27 | 1.10–1.41 | 1.04–1.31 | | 0.89–1.19 | 0.96–1.30 | 0.91–1.19 | 0.97–1.26 | 0.97–1.26 | |
| **Age 14–15** | | | | | | | | | | | | | | | | | | |
| 1–3 (poorest) | 11.2 | 11.5 | 12.3 | 12.7 | 13.4 | | 12.9 | 13.2 | 14.0 | 14.8 | 15.9 | | 9.4 | 9.8 | 10.4 | 10.4 | 10.7 | |
| 4–5 | 11.7 | 12.1 | 12.6 | 13.0 | 13.7 | | 13.1 | 13.6 | 14.1 | 14.4 | 15.3 | | 10.3 | 10.5 | 11.1 | 11.5 | 11.9 | |
| 6–7 | 9.2 | 9.8 | 10.6 | 10.8 | 11.5 | | 10.3 | 10.7 | 11.4 | 11.8 | 12.5 | | 8.1 | 8.9 | 9.6 | 9.8 | 10.4 | |
| 8–10 (wealthiest) | 6.7 | 6.8 | 7.5 | 7.6 | 8.1 | | 7.5 | 7.5 | 8.3 | 8.1 | 8.5 | | 5.8 | 6.1 | 6.7 | 7.0 | 7.5 | |
| Rate Ratio Poorest/ Wealthiest | 1.68 | 1.69 | 1.63 | 1.68 | 1.66 | 0.59 | 1.72 | 1.75 | 1.68 | 1.82 | 1.86 | 0.07 | 1.61 | 1.60 | 1.56 | 1.50 | 1.42 | 0.004 |
| 99% CI | 1.58–1.79 | 1.59–1.80 | 1.55–1.72 | 1.59–1.77 | 1.57–1.74 | | 1.59–1.87 | 1.62–1.90 | 1.57–1.81 | 1.70–1.96 | 1.74–1.99 | | 1.46–1.78 | 1.49–1.76 | 1.43–1.70 | 1.38–1.62 | 1.31–1.53 | |

Finally, when interpreting this study, it is important to keep in mind that each year's data represents a cross-section of the Israeli population, and that the same individuals are not tracked from year to year. This has to do with the mechanism of collecting the data, as explained above. As such, our results should not be interpreted as saying that a group of Israeli children, followed across the life course, were gaining weight rapidly from year to year. Although this may have happened, our data cannot show this. Rather, our data should be interpreted as showing that successive age cohorts during the COVID period had higher BMI each year than the year before.

# 5. Conclusions

In conclusion, we present rates of underweight, normal weight, overweight, and obese, across a five-year period from 2017–2021, representing the entire population of Israel. These weight distributions were captured at ages 7 and 14–15. We also present the percentage of children with obesity over time stratified by area-level SES. Our results show that the rates of overweight and obesity, which had been steady or improving between 2017–2019, increased during 2020 and 2021 –presumably due to lifestyle changes associated with the COVID-19 epidemic. SES- and sex-stratified analyses suggested that socioeconomic disparities in obesity at age 14–15 are worse among boys, and in fact worsened still further during the COVID period, while disparities among girls of this age lessened somewhat. Our study adds to the international literature about the impact of the COVID-19 period on overweight and obesity among children. In our local context, these findings call for us to redouble our efforts to encourage healthier lifestyles among our children, as an investment in their long-term health.

## Supporting information

**S1 Checklist. STROBE statement—checklist of items that should be included in reports of observational studies.**
(DOCX)

## Acknowledgments

The authors wish to thank the members of the QICH Steering Committee: Dr. Gabriella Lawrence, Dr. Shuli Brammli-Greenberg, Dr. Ehud Horowitz, and Prof. Orly Manor. In addition, the authors wish to thank the HMOs and participants in the steering committee meetings, and the Israel National Institute for Health Policy Research for supporting the QICH program.

## Author Contributions

**Conceptualization:** Adam Rose, Ronit Calderon-Margalit.

**Data curation:** Eliana Ein Mor, Michal Krieger, Arnon D. Cohen, Eran Matz, Edna Bar-Ratson, Ronen Bareket, Ronit Calderon-Margalit.

**Formal analysis:** Eliana Ein Mor.

**Investigation:** Michal Krieger, Arie Ben-Yehuda.

**Methodology:** Adam Rose, Shoshana Revel-Vilk.

**Project administration:** Eliana Ein Mor.

**Resources:** Shoshana Revel-Vilk, Arnon D. Cohen, Eran Matz, Edna Bar-Ratson, Ronen Bareket, Ronit Calderon-Margalit.

**Supervision:** Arie Ben-Yehuda, Ora Paltiel, Ronit Calderon-Margalit.

**Writing – original draft:** Adam Rose.

**Writing – review & editing:** Adam Rose, Eliana Ein Mor, Michal Krieger, Arie Ben-Yehuda, Shoshana Revel-Vilk, Arnon D. Cohen, Eran Matz, Edna Bar-Ratson, Ronen Bareket, Ora Paltiel, Ronit Calderon-Margalit.

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
