## [Decision Letter · Decision Letter 0]

2 Jul 2023

PONE-D-23-13315Pediatric Overweight and Obesity Increased in Israel During the COVID-19 PeriodPLOS ONE

Dear Dr. Rose,

Thank you for submitting your manuscript to PLOS ONE. After careful consideration, we feel that it has merit but does not fully meet PLOS ONE’s publication criteria as it currently stands. Therefore, we invite you to submit a revised version of the manuscript that addresses the points raised during the review process.

There are some serious methodological issues addressed by one of the reviewers. Please have a look at the comments and revise where needed.

We look forward to receiving your revised manuscript.

Kind regards,

Inge Roggen, M.D., Ph.D.

Academic Editor

PLOS ONE

Reviewers' comments:

Reviewer's Responses to Questions

**Comments to the Author**

1. Is the manuscript technically sound, and do the data support the conclusions?

Reviewer #1: Partly

Reviewer #2: Yes

2. Has the statistical analysis been performed appropriately and rigorously? 

Reviewer #1: Yes

Reviewer #2: Yes

3. Have the authors made all data underlying the findings in their manuscript fully available?

Reviewer #1: Yes

Reviewer #2: Yes

4. Is the manuscript presented in an intelligible fashion and written in standard English?

Reviewer #1: No

Reviewer #2: Yes

5. Review Comments to the Author

Reviewer #1: Dear authors, your manuscript fails to clarify;

1. The difference between census and survey

2. compares with the data before the occurrence of COVID 19

3. Establishing research territory

4.Is the factor related to obesity occur just in ONE year?

5. The language, grammar and complete construction of sentence and paragraphs

6.You have used confidence interval. Hence, is that sampled or per the entire population?

7. Appropriate presentation of the result, discussion, conclusion and recommendations as well as reference sections!

Reviewer #2: Globally, overweight and obesity are serious public healthcare concerns. The recent outbreak of Covid-19 has revealed vulnerabilities in most health systems and socioeconomic disparities among the world population. Thank you for your efforts in making this significant contribution to literature. However, I have some minor remarks in your discussion section.

"Our study had important strengths." This is an overstatement. Only a single strength has been stated. Where are the other strengths? Please state the other strengths.

"Another point in support of these data being real is that the documentation of height and weight decreased much more markedly among 14-15-year-olds than among 7-year-olds, and yet similar increases in obesity were observed in both age groups."

This statement may be construed as if there is doubt about the authenticity of the study data. Could you reconsider structuring this statement?

6. PLOS authors have the option to publish the peer review history of their article (what does this mean?). If published, this will include your full peer review and any attached files.

Reviewer #1: No

Reviewer #2: **Yes: **Dr. Agnes Arrey

---

## [Author Response · Author response to Decision Letter 0]

22 Jul 2023

Please see attached document, "Response to Reviewers"

---

## [Editor Report · Decision Letter 1]

21 Aug 2023

Pediatric Overweight and Obesity Increased in Israel During the COVID-19 Period

PONE-D-23-13315R1

Dear Dr. Rose,

We’re pleased to inform you that your manuscript has been judged scientifically suitable for publication and will be formally accepted for publication once it meets all outstanding technical requirements.

Kind regards,

Inge Roggen, M.D., Ph.D.

Academic Editor

PLOS ONE
---

## [Editor Report · Acceptance letter]

24 Aug 2023

PONE-D-23-13315R1 

Pediatric overweight and obesity increased in Israel during the COVID-19 period 

Dear Dr. Rose:

I'm pleased to inform you that your manuscript has been deemed suitable for publication in PLOS ONE. Congratulations! Your manuscript is now with our production department. 

Kind regards, 

on behalf of

Prof. Inge Roggen 

Academic Editor

PLOS ONE